# Endophytic Bacterium *Flexivirga meconopsidis* sp. nov. with Plant Growth-Promoting Function, Isolated from the Seeds of *Meconopsis integrifolia*

**DOI:** 10.3390/microorganisms11122899

**Published:** 2023-11-30

**Authors:** Yongtao Kan, Li Zhang, Yan Wang, Qingyun Ma, Yiqing Zhou, Xu Jiang, Wei Zhang, Zhiyong Ruan

**Affiliations:** 1College of Life Sciences, Xinjiang Normal University, Urumqi 830017, China; kyt3117428630@126.com; 2State Key Laboratory of Efficient Utilization of Arid and Semi-Arid Arable Land in Northern China, Institute of Agricultural Resources and Regional Planning, Chinese Academy of Agricultural Sciences, Beijing 100081, China; a18301335070@163.com (Y.W.); mqy@webmail.hzau.edu.cn (Q.M.); zhouyiqing@caas.cn (Y.Z.); jiangxu@caas.cn (X.J.); 3CAAS-CIAT Joint Laboratory in Advanced Technologies for Sustainable Agriculture, Institute of Agricultural Resources and Regional Planning, Chinese Academy of Agricultural Sciences, Beijing 100081, China; 4College of Life Sciences, Yantai University, Yantai 264005, China; wkiaazl@outlook.com; 5College of Resources and Environment, Tibet Agricultural and Animal Husbandry University, Linzhi 860000, China; 6National Key Laboratory of Agricultural Microbiology, Huazhong Agricultural University, Wuhan 430070, China

**Keywords:** *Flexivirga meconopsidis*, *Meconopsis integrifolia*, genome, plant growth promotion, shotgun proteomics

## Abstract

Strain Q11^T^ of an irregular coccoid Gram-positive bacterium, aerobic and non-motile, was isolated from *Meconopsis integrifolia* seeds. Strain Q11^T^ grew optimally in 1% (*w*/*v*) NaCl, pH 7, at 30 °C. Strain Q11^T^ is most closely related to *Flexivirga*, as evidenced by 16S rRNA gene analysis, and shares the highest similarity with *Flexivirga aerilata* ID2601S^T^ (99.24%). Based on genome sequence analysis, the average nucleotide identity and digital DNA–DNA hybridization values of strains Q11^T^ and D2601S^T^ were 88.82% and 36.20%, respectively. Additionally, strain Q11^T^ showed the abilities of nitrogen fixation and indole acetic acid production and was shown to promote maize growth under laboratory conditions. Its genome contains antibiotic resistance genes (the vanY gene in the vanB cluster and the vanW gene in the vanI cluster) and extreme environment tolerance genes (ectoine biosynthetic gene cluster). Shotgun proteomics also detected antibiotic resistance proteins (class A beta-lactamases, D-alanine ligase family proteins) and proteins that improve plant cold tolerance (multispecies cold shock proteins). Strain Q11^T^ was determined to be a novel species of the genus *Flexivirga*, for which the name *Flexivirga meconopsidis* sp. nov. is proposed. The strain type is Q11^T^ (GDMCC 1.3002^T^ = JCM 36020 ^T^).

## 1. Introduction

The genus *Flexivirga*, a member of the Phylum *Actinobacteria,* was first described by Anzai et al. [1]. The List of Prokaryotic Names with Standing in Nomenclature (https://lpsn.dsmz.de/search?word=Flexivirga, accessed on 8 October 2023) contains six species with legitimately recognized names, i.e., *Flexivirga alba* [1], *Flexivirga endophytica* [2], *Flexivirga lutea* [3], *Flexivirga oryzae* [4], *Flexivirga caeni* [5], and *Flexivirga aerilata* [6], which were isolated from the leaves of basil [2], the feces of ibis [3], the soil of a rice paddy [4], activated sludge [5], and an automobile air conditioning system [6], respectively. All members of the genus *Flexivirga* are described as Gram-positive bacteria, aerobic, non-motile, and with a high G + C content in their DNA (~67 mol%).

*Meconopsis integrifolia* is mostly distributed in the Tibetan Plateau. Many researchers found that this medicinal plant contains large amounts of bioactive compounds such as flavonoids and alkaloids, which are used in the pharmaceutical and healthcare industries [7]. However, the survival and growth of *Meconopsis integrifolia* have been significantly challenged by environmental changes, habitat loss, overexploitation, and difficulties in artificial cultivation [8].

Endophytic bacteria are symbiotic and beneficial organisms that colonize plants without causing disease; they are also plant growth-promoting bacteria (PGPB) [9]. PGPB promote plant growth and development and improve plant adaptability to the environment by secreting various hormones, facilitating the uptake of dissolved minerals, and carrying out nitrogen fixation [10,11]. Some endophytes, such as *Paenibacillus polymyxa* SK1 [12], *Bacillus cereus* N5 [13], *Burkholderia phytofirmans* PsJN [14], along with *Kocuria* sp. TRI2-1, *Micromonospora* sp. TSI4-1, and *Sphingomonas* sp. MG-2 [15], exhibit the ability to fix nitrogen, dissolve phosphate, produce IAA, and promote plant growth.

Therefore, this study aimed to characterize the new species *Flexivirga meconopsidis* Q11^T^, which inhabits the seeds of the medicinal plant *Meconopsis integrifolia* from the Qinghai–Tibet Plateau, by genome and proteomics analysis. The growth-promoting ability of strain Q11^T^ was identified by analyzing IAA production, phosphorus dissolution, and nitrogen fixation, and the effect of strain Q11^T^ on plant growth was further confirmed by pot experiments. As far as we know, this is the first report of a species of the genus *Flexivirga* that can promote plant growth.

## 2. Materials and Methods

### 2.1. Isolation and Cultivation

Strain Q11^T^ was isolated from the seeds of *Meconopsis integrifolia* using Beef Extract Peptone Agar medium (BA). The surface of the seeds was disinfected [16], followed by crushing the seeds in a ceramic bowl and diluting them with sterile water. Each diluted sample (100 µL) was separately spread on BA medium. After being cultured at 30 °C for 3 days, a white colony Q11^T^ was isolated and purified. Strain Q11^T^ was stored at −80 °C in sterile 60% (*v*/*v*) glycerol.

### 2.2. 16S rRNA Gene Phylogeny

We performed 16S rRNA gene amplification using the process outlined by Li et al. [17]. The sequence was determined by the Life Technologies Company (Shenzhen, China). The sequence was uploaded to the EzBioCloud database for identification. By using the CLUSTAL_W program in MEGA (Version 7.0) software for sequence alignment, the neighbor-joining (NJ) [18], maximum likelihood (ML) [19], and maximum parsimony (MP) [20] methods were applied to analyze the genetic relationship among the strains, and a phylogenetic tree was constructed [21]. By performing 1000 repeated bootstrap analyses, the phylogenetic distance was calculated using the Kimura two-parameter model [22].

### 2.3. Genome Features

The genomic DNA of the strain Q11^T^ was collected [23]. The genome sequence was completed on the Illumina MiSeq platform by Guangzhou Magi company (Guangzhou, China). The obtained sequence was submitted to NCBI, and we obtained the genome number JAOBQJ000000000. The average nucleotide identity (ANI) and digital DNA–DNA hybridization (dDDH) values of strain Q11^T^ and its closest members (*F. aerilata* ID2601S^T^, *F. caeni* BO-16^T^, *F. endophytica* KCTC 39536^T^, and *F. oryzae* R1^T^) were compared by OrthoANIu and the genome-to-genome distance calculator. In addition, gene analysis and annotation were carried out by the RAST server (version 2.0) [24] and Proksee [25]. The gene clusters of secondary metabolism biosynthesis in bacterial genome were analyzed by the antiSMASH program (version 7.0.1) [26,27].

### 2.4. Physiology

The morphology of strain Q11^T^ was observed by transmission electron microscopy (Hitachi 7500) [28]. Strain Q11^T^ was cultured under various conditions using BA to examine its physiological properties. Strain Q11^T^ was subjected to Gram staining [29]. Cell motility was observed by the semi-solid puncture method [30]. To determine the culture conditions for the growth and development of the strain, the growth status of the strain was observed at different temperatures (10–50 °C), different pH values (pH = 4.0–11.0), and different NaCl concentrations (0, 0.5, and 1–10% at intervals of 1.0%, *w*/*v*) [31]. The pH value was adjusted in BA liquid medium [32]. Several key characteristics were studied according to the bacterial identification manual [33], using the methyl red test and the Voges–Proskauer test and determining the occurrence of catalases and oxidases and the hydrolysis of Tween 20, 40, 60, and 80 (final concentration 1%), starch (0.2%, *w*/*v*), and gelatin (10%, *w*/*v*). Further physiological and biochemical properties of the cells were identified by cultivating them at 30 °C and using the API ZYM, 20NE, and 50CH test strips (bioMérieux) as described by the manufacturer.

### 2.5. Chemotaxonomy

Well-grown cells of strain Q11^T^ cultured on BA liquid medium for 24 h at 30 °C were used for fatty acid analysis. The fatty acids were identified by the Sherlock microbial identification system (MIDI) and GC (6890 N, Agilent), and the peaks were identified using the RTSBA6 database [34]. Respiratory quinones of strain LT1^T^ were extracted from freeze-dried cells [35] and analyzed by LC-MS (Agilent 1260) [36]. Polar lipids were extracted from freeze-dried bacteria [37] and separated by two-dimensional TLC (MERCK, Silica gel 60) [36]. Total lipids were measured in a molybdophosphoric acid hydrate ethanol solution, amino lipids using the ninhydrin reagent, phospholipids using the Zinzadze reagent, and glycolipids using the α-naphthol reagent [38].

### 2.6. Shotgun Proteomics Analyses

The global profile of the protein/polypeptide complement in strain Q11^T^ cells and supernatants was detected by shotgun proteomics [39,40]. The cells were washed with sterile water 3 times, frozen in liquid nitrogen for 10 min, and transported to Shanghai Paisenuo Biology Co., Ltd. (Shanghai, China) at −80 °C for detection. The samples were analyzed by an ORBITRAP ECLIPSE mass spectrometer (Thermo Scientific, Waltham, MA, USA) and a FAIMSPro™ Interface instrument (Thermo Scientific) using the NanosprayFlex™ (NSI, London, UK) ion source and setting the ion spray voltage to 2.0 kV, in the data-dependent acquisition mode, and the full scanning range was *m*/*z* 350–1500. The data were obtained by searching the database with Proteome Discoverer 2.4 software.

### 2.7. Determination of the Growth-Promoting Ability

The ability of strain Q11^T^ to produce indole acetic acid (IAA) was evaluated [41], and the yield of IAA was quantitatively at the 535 nm wavelength. The nitrogen fixation ability of strain Q11^T^ was evaluated [42], and the nitrogenase activity of the strain was determined by a nitrogenase enzyme-linked immunoassay kit (LMAI Bio, Shanghai, China). Chrome Azurol S (CAS) agar medium was used to determine whether strain Q11^T^ had the ability to produce siderophores [43]. Strain Q11^T^ was inoculated in organophosphorus (Mongina medium, Hopebiol, Qingdao, China) and inorganic phosphorous media and observed for the appearance of a phosphorus solubilization halo around the cells [44].

### 2.8. Potting Test

Strain Q11^T^ was inoculated in LB liquid medium for 24 h, washed three times with sterile water (centrifuged at 8000 r/min for 5 min), and finally diluted with sterile water to OD_600_ = 1. Based on existing research [8], the potting experiment is difficult to perform due to the obstacles posed by the germination of *Meconopsis integrifolia*; so, we chose corn seeds for the plant growth promotion experiments (Cultivar *Zhengdan* 958). We used black soil, pH 6.01 ± 0.11, with an organic matter content of 26.45 ± 0.86 g/kg; other soil properties are shown in Appendix A. Surface-disinfected maize seeds were sown into pots (bottom, 5 cm, top, 7.5 cm, height, 10.5 cm) containing 200 g of soil [45], placing 3 seeds per pot. The control group (CK) received 10 mL of sterile water. In the experimental group (TG), we applied 10 mL of bacterial solution. Plant height and root length (cm), stem and root fresh weight (g), stem and root dry weight (g), stem thickness (cm), and relative leaf chlorophyll content were measured after 2 weeks [45,46]. The relative content of chlorophyll (SPAD) was measured by a handheld chlorophyll meter (Konica Minolta SPAD-502 Plus, Tokyo, Japan) [47,48].

The GenBank accession numbers for the 16S rRNA gene sequence and draft genome sequences are OR016664 and JAOBQJ000000000, respectively.

RAST data: https://rast.nmpdr.org/seedviewer.cgi?page=Organism&organism=2977121.4 (accessed on 14 November 2023); antiSMASH data: https://antismash.secondarymetabolites.org/upload/bacteria-0f8777f2-48fc-4834-ac26-cad8c79c868e/index.html (accessed on 20 October 2023).

## 3. Results

### 3.1. Phylogenetic Analysis

The 16S rRNA gene sequence obtained from the analysis of the genomic DNA of strain Q11^T^ was found to comprise 1524 bp (accession number OR016664). After comparing and querying the gene sequence of this strain in EzBioCloud and Blast, it was observed that the similarity between this strain and strains of species of *Flexivirga* was the highest. The strain had high sequence similarity with *F. aerilata* ID2601S^T^ (99.24%), *F. caeni* BO-16^T^ (97.64%), *F. endophytica* KCTC 39536^T^ (97.30%), *F. alba* ST13^T^ (97.09%), *F. oryzae* R1^T^ (97.02%), and *F. lutea* KCTC 39625^T^ (96.31%). Phylogenetic analysis based on the NJ, MP, and ML methods revealed that strain Q11^T^ clustered together with members of the genus *Flexivirga* in the phylogenetic tree (Figure 1, Appendix A). This indicated that strain Q11^T^ belongs to the genus *Flexivirga*.

### 3.2. Genome Sequence Analysis

The genome size of strain Q11^T^ was found to be 4.28 Mbp. The genome appeared to contain 10 contigs with an N50 value of 750,316 coding sequences. We found that the G + C content of the DNA was 68.47 mol%, the protein-coding genes were 4144, and the average gene length was 959.06, and there were 50 tRNAs and 4 rRNAs. The ANI and dDDH values of strain Q11 ^T^ and strains *F. aerilata* ID2601S^T^, *F. caeni* BO-16^T^, *F. Endophytica* KCTC 39536^T^, and *F. oryzae* R1^T^ were in the 77.24–88.82% and 21.00–36.20% ranges (Appendix A), respectively, and were lower than the threshold values of these spccies [49].

The RAST annotation of the strain Q11^T^ genome showed 281 subsystems, with a coverage rate of 24%. These subsystems mainly include amino acids and derivatives, carbohydrates, and protein metabolism (Appendix A). The genome of strain Q11^T^ was annotated using Proksee. Antibiotic resistance gene predictions indicated that there were two antibiotic resistance gene clusters. These clusters include the vanY gene in the vanB cluster [50] and the vanW gene in the vanI cluster [51]. Both the vanW and the vanY genes contribute to the gene cluster encoding glycopeptide antibiotics with a resistance mechanism associated with antibiotic target alteration [52,53]. The results of CRISPRCasFinder Annotation analysis showed that strain Q11^T^ had five CRISPR arrays and no CAS array (Figure 2).

The biosynthesis gene clusters were analyzed on the antiSMASH webpage, and the results are shown in Table 1. In Region 1, we found four types of protein synthesis gene clusters. The first was the terpene gene cluster, which is involved in the first step of the production of terpenoid compounds, which can be processed into perfumes, plant hormones, drugs, etc. [54]. It showed 38% similarity to the hopene biosynthetic gene cluster, which is involved in the synthesis of bioactive hopanoids that contribute to bacterial stress resistance [55]. The second included the non-alpha poly-amino acid (NAPAA) and RIPP-like biosynthetic gene clusters, showing 100% similarity to known gene clusters for the biosynthesis ε-poly-L-lysine, which mediates resistance to pathogenic bacteria [56]. The third was the proteusin gene cluster, which encodes ribosomally synthesized and post-translationally modified peptide (RiPP) natural products [57] and has a similarity of 14% to the aborycin biosynthetic gene cluster, which is involved in aborycin production [58]. The fourth was the betalactone gene cluster, which is involved in the production of natural β-lactone [59]. In Region 2, three types of protein synthesis gene clusters were identified. The first was also a terpene gene cluster. The second was the phenazine gene cluster, showing 30% similarity with the 5-acetyl-5,10-dihydrophenazine-1-carboxylic acid biosynthetic gene cluster; phenazine mediates resistance to pathogenic bacteria [60]. The third was the NI–siderophore gene cluster, showing 75% similarity to the desferrioxamine E biosynthetic gene cluster [61]. In region 3, there were two types of protein synthesis gene clusters. One was the linaridin gene cluster, showing 33% similarity to the 5-dimethylallylindole-3-acetonitrile gene cluster involved in a new pathway of tryptophan metabolism [62], probably requiring the biosynthesis of isoprenyl indole derivatives. The other was the ectoine gene cluster, showing 8% similarity to the kosinostatin biosynthetic gene cluster, kosinostatin being an antitumor antibiotic [63]. In region 6, only the ectoine gene cluster was predicted. Previous research [64] indicated that both kosinostatin and ectoine could protect cells from the pressure caused by the external environment.

### 3.3. Physiology and Chemotaxonomy

After 3 days of incubation on BA plates at 30 °C, the strain Q11^T^ colonies had acquired slightly irregular shapes and were light brown, rod-shaped (0.67–0.77 μm in width and 0.77–0.96 μm in length) (Appendix A), and non-motile. The temperature, pH, and NaCl concentration for strain Q11^T^ culture were 25–40 °C, 6.0–8.0, and 0–4.0%, respectively. Characteristic differences between strain Q11^T^ and the reference strains are shown in Table 2.

The cellular fatty acids (>5%) of strain Q11^T^ were summed feature 9 (C_16:0_ 10-methyl and/or iso-C_17:1_ *ω*9c) (22.5%), iso-C_16:0_ (19.7%), iso-C_17:0_ (10.5%), anteiso-C_17:0_ (8.7%), C_18:0_ (7.3%), C_18:0_ 10-methyl (7.0%), and other types (>0.5%), as shown in Table 3. MK-8 (H4) and MK-8 (H6) were the most abundant isoprenoid quinones. The major polar lipids in strain Q11^T^ were diphosphatidylglycerol (DPG), two aminophospholipids (APL1-2), ten phospholipids (PL1-10), two aminolipids (AL1-2), and two glycolipids (GL1-2) (Appendix A).

### 3.4. Shotgun Proteomic Analysis

The protein results of strain Q11^T^ are shown in Appendix A. It shows that 245 proteins and 794 peptides were identified in the supernatant, and 2401 proteins and 20,413 peptides were identified in the cells (Appendix A).

Siderophore-interacting proteins (WP_265447369.1, WP_265443731.1) were detected in the cells [65]. These proteins can interact with other proteins to obtain elemental iron from the environment and also possess their own functions [66,67]. The adenosyl-hopene transferase HpnH (WP_265442571.1) was detected in the cells. Adenosylhopane is a precursor of C_35_ hopanoids, which can regulate bacterial cell membrane fluidity and permeability [68] and also helps promote beneficial interactions between bacteria and plants [69]. Acetyl-CoA C-acetyltransferase (WP_265443840.1, WP_265444891.1, WP_265445749.1) was detected in both cells and supernatant. It is used to synthesize terpenoids in *Platycodon grandifloras*, which is known as a traditional Asian herbal medicine and functional food [70]. The enzyme 1-deoxy-D-xylulose-5-phosphate synthase (WP_279671918.1) was detected in the cells; it is involved in the synthesis of terpenes, which are volatile substances with a unique fragrance that attract pollinators and protect against pests and diseases in plants [71]. The gentamicin biosynthesis-related Gfo/Idh/MocA oxidoreductase family (WP_265446697.1) [72] and the cobalamin biosynthesis-related cobaltochelatase subunit CobN (WP_265447325.1) [73] were also detected in the cells. More importantly, the multispecies cold shock protein (WP_171152712.1) was detected in both cells and supernatant. Studies pointed out that this protein can induce frost tolerance in plants in a low-temperature environment without affecting their normal growth and development [74].

### 3.5. IAA Production and Nitrogen Fixation Ability of Strain Q11^T^

Strain Q11^T^ could produce IAA. The IAA content per unit volume of fermentation broth was 3.26 mg/L. (Appendix A and Appendix A). Strain Q11^T^ showed the ability of nitrogen fixation, and its nitrogenase activity was 181.80 IU/L (Appendix A and Appendix A). IAA promotes cell division and differentiation, seed germination, and plant growth [75]. Biological nitrogen fixation is an important source of active nitrogen in the ecosystem [76]. It can not only increase soil nutrients but also improve plant habitat [77] and quickly establish a stable community suitable to plant growth [78].

#### Effects on Maize Growth and Development

Compared with CK, the plant height, root length, fresh weight of stem and root, and dry weight and thickness of stem and root of maize were increased significantly in the experimental group (Figure 3). There were significant differences between TG and CK in root dry weight and stem thickness, plant height, root length, fresh weight of stem and root, and stem dry weight. The experimental treatment did not affect the chlorophyll content of the leaves. The results showed that the inoculation of strain Q11^T^ had a positive effect on the growth and development of maize.

## 4. Description of *Flexivirga meconopsidis* sp. nov.

### Flexivirga meconopsidis (me.co.nop’si.dis. N.L. gen. n. Meconopsidis, of the Plant Meconopsis integrifolia)

The strain is characterized by aerobic, Gram-positive, rod-shaped, non-motile, irregular coccoid cells, 0.67–0.77 μm in width and 0.77–0.96 μm in length. The most suitable conditions for strain growth were 30 °C, pH 7.0, and 1% NaCl concentration. The cells were found to be oxidase-negative and catalase-positive. They showed ability to hydrolyze Tween 60. The API ZYM strip analysis showed positive results for alkaline phosphatase 8, lipase (C14), leucine arylamidase, trypsin, acid phosphatase, naphthol-AS-BI-phosphohydrolase, α-galactosidase, β-galactosidase, α-glucosidase, β-glucosidase, N-acetyl-β-glucosaminidase, α-mannosidase, and β-fucosidase, and negative results for other enzymes. The results for gelatinase, β-galactosidase, glucose, mannose, N-acetyl-glucosamine, maltose, gluconate, malic acid, and citrate of strain Q11^T^ were positive, but the results for other indicators of assimilation were negative in the API 20NE test strip. The API 50CH strip test showed positive results for erythritol, D-galactose, D-glucose, D-fructose, D-mannose, potassium gluconate, and 2-ketogluconate, whereas other results of this test were negative. The major fatty acids (>10%) found were summed feature 9 (C_16:0_ 10-methyl and/or iso-C_17:1_ *ω*9c), iso-C_16:0_, and iso-C_17:0_. The major respiratory quinones were MK-8 (H4) and MK-8 (H6). The main polar lipids were DPG, APL, PL, AL, and GL.

The strain was designated Q11^T^ (=GDMCC 1.3002^T^=JCM 36020^T^); it was isolated from the seeds of *Meconopsis integrifolia*. The genomic DNA G + C content of this strain is 68.47 mol%.

## 5. Discussion

Endophytic microorganisms are often closely related to plant physiological metabolism in a reciprocal manner [10,11]. In this study, we isolated the new species Q11^T^ from *Meconopsis integrifolia* seeds. Genomic ANI and dDDH analyses, together with physiological and biochemical analyses, confirmed that strain Q11^T^ is a novel species of the genus *Flexivirga*. Compared with the reported strains of the genus *Flexivirga* [1,2,3,4,5,6], strain Q11^T^ has the abilities of nitrogen fixation (181.80 IU/L), IAA production (3.26 mg/L in 2 days), and plant growth promotion, this last function being described for the first time for the genus *Flexivirga*. Previous research showed that endophytes promote plant growth through nitrogen fixation, phosphorus dissolution, and growth hormone production [12,79]. Endophytic bacteria with growth-promoting effects were isolated from plants such as rice [45], maize [80], black pepper [81], *Arctium lappa L.* [79], and *Camellia sinensis* [82]. Therefore, the isolated strain Q11^T^ has important application potential in plant growth and agriculture.

Antibiotic resistance genes are the root cause of bacterial drug resistance. Various types of antibiotic resistance genes have been found, which cause environmental pollution [83]. They can be transmitted through bird feces [83]. Glycopeptide resistance genes (the vanY gene in the vanB cluster [50] and the vanW gene in the vanI cluster [51]) were found in strain Q11^T^ isolated from the seeds of *Meconopsis integrifolia* grown in alpine areas. The secondary-metabolite synthesis gene cluster has significant value in a wide range of fields, such as drug discovery [84] and biological protection [62]. In this study, IAA-related genes (5-dimethylallylindole-3-acetonitrile [62]) and extreme environment tolerance genes (ectoine biosynthetic gene cluster [64]) were predicted in strain Q11^T^, which were related to the habitat of the host *Meconopsis integrifolia*.

Proteomics is a method to study the interactions, functions, composition, structures, and cellular activities of proteins. Proteomics results reflects cell function better than those from genomic studies [85]. Proteomics also partially confirmed the genomic predictions that the new strain contains antibiotic resistance proteins (class A beta-lactamase [86], D-alanine ligase family protein [87]), siderophore transport proteins (siderophore-interacting protein [65]), a protein that improves plant cold tolerance (multispecies cold shock protein [74]), and some proteins involved in terpenoid synthesis (adenosyl-hopene transferase HpnH [68], acetyl-CoA C-acetyltransferase [70], and 1-deoxy-D-xylulose-5-phosphate synthase [71]). Microorganisms use iron carriers to provide plants with this metal, which stimulates growth and increases plant resistance to stress [88,89]. It was reported that endophytes obtain nutrients more easily than other microorganisms [90]. Cold shock proteins could help bacteria not only withstand freezing temperatures [90] but also grow in cold environments [91]. They are mainly isolated from bacteria living in extreme environments such as glaciers [92] and cold ecosystems [93]. Therefore, it is speculated that the production of the identified cold shock protein could make the new strain tolerate extreme environments and further ensure the growth and development of the host *Meconopsis integrifolia* seeds at low temperatures.

Previous studies showed that the growth-promoting characteristics of the known strains are related to many mechanisms such as the presence of iron carriers, IAA production, and nitrogen fixation [12,79]. This study also confirmed through various experiments that the isolated strain Q11^T^ has the ability to promote plant growth.

## 6. Conclusions

In conclusion, this study showed that strain Q11^T^ isolated from the seeds of *Meconopsis integrifolia* is a novel species of the genus *Flexivirga*. Genomic analysis showed that it contains antibiotic resistance genes, IAA production-related genes, and extreme environment tolerance genes. The proteomic results showed that the strain produces antibiotic resistance proteins, iron transport proteins, a plant cold tolerance protein, and terpenoid synthesis proteins. The results of the functional experiments showed that the strain has the ability to produce IAA, fix nitrogen, and promote plant growth. This study is also the first to report that species of the *Flexivirga* genus have the function of promoting plant growth and provides some insights for improving plant growth and development.

## Figures and Tables

**Figure 1 microorganisms-11-02899-f001:**
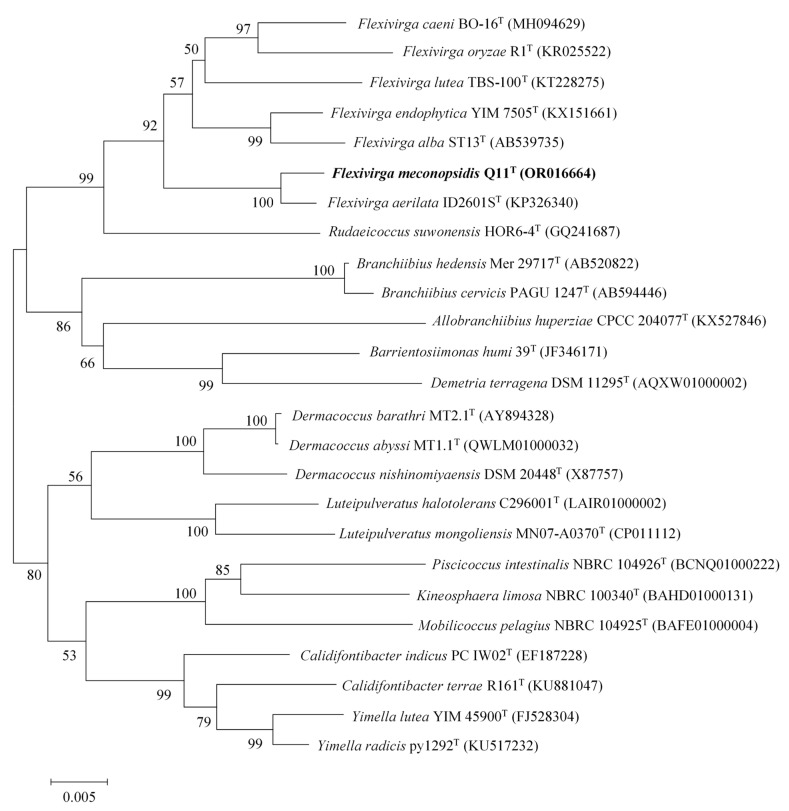
Neighbor-joining phylogenetic tree of the 16S rRNA gene sequence showing the relationship between strain Q11^T^ and strains of species of the genus *Flexivirga*. The number of bootstrap replications was 1000. Shown in parentheses are the accession numbers of the 16S rRNA sequences in the NCBI database. Bar, 0.005 nucleotide substitutions per position.

**Figure 2 microorganisms-11-02899-f002:**
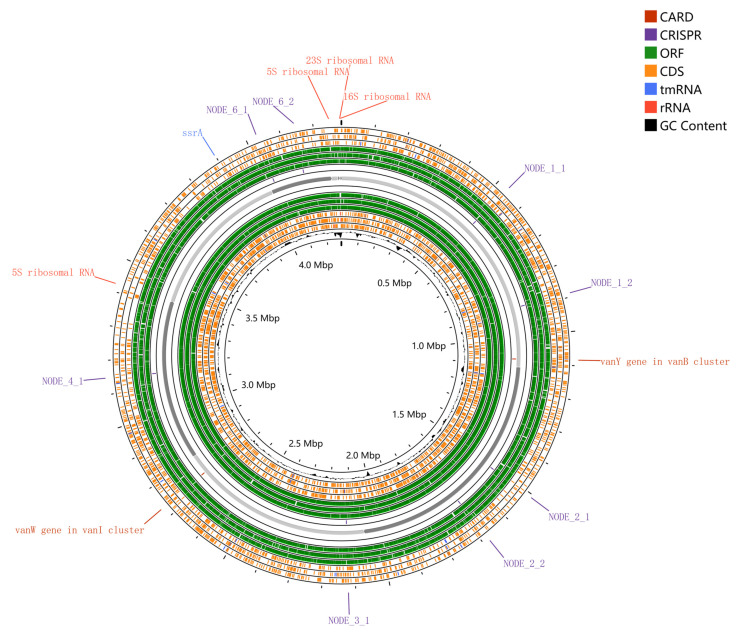
Annotation of the genome of strain Q11^T^ through Proksee. CARD: comprehensive antibiotic resistance database, CRISPER: CRISPR arrays, ORF: open reading frames, CDS: coding sequence.

**Figure 3 microorganisms-11-02899-f003:**
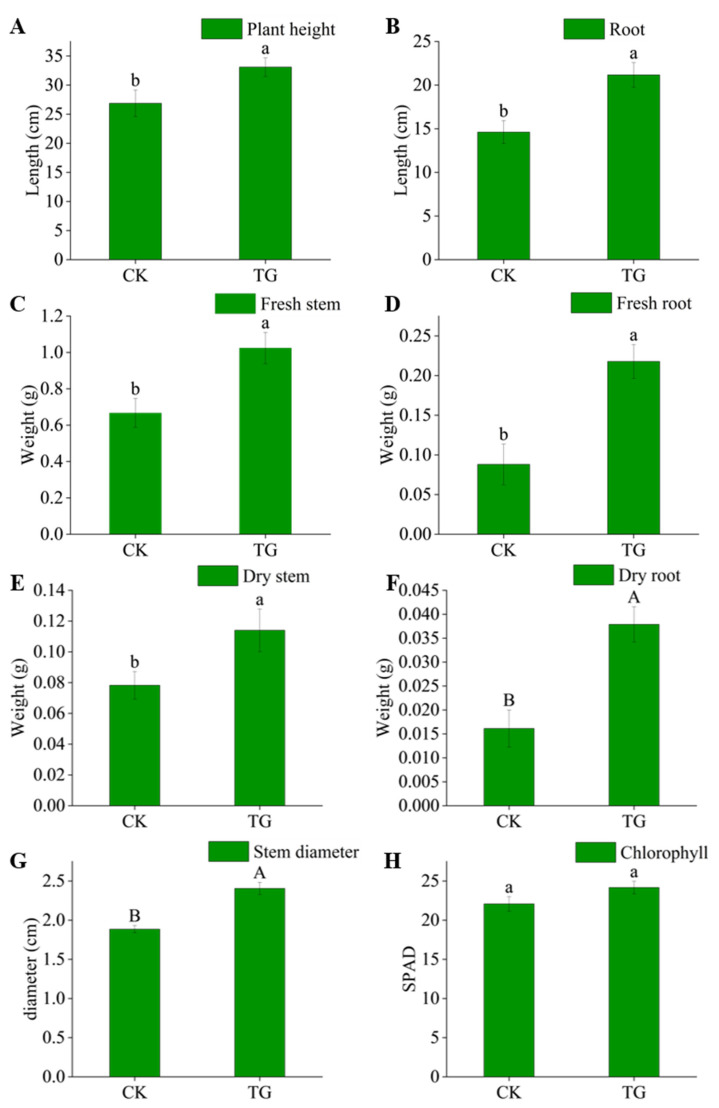
Effect of strain Q11^T^ on the growth of maize. Plant height (**A**), root length (**B**), fresh stem weight (**C**), fresh root weight (**D**), dry stem weight (**E**), dry root weight (**F**), stem diameter (**G**), and chlorophyll content (**H**). Different letters indicate significant differences between treatments (a, b for *p* < 0.05, A, B for *p* < 0.01).

**Table 1 microorganisms-11-02899-t001:** The distribution of biosynthetic gene clusters in strain Q11^T^ determined using antiSMASH.

Region	Type	From-To	Most Similar Known Cluster	Similarity
Region 1.1	terpene	133,677–159,753	hopene	38%
Region 1.2	NAPAA, RiPP-like	412,322–454,746	ε-poly-L-lysine	100%
Region 1.3	proteusin	650,410–671,897	aborycin	14%
Region 1.4	betalactone	798,229–827,406		
Region 2.1	terpene	131,910–152,806		
Region 2.2	phenazine	682,797–708,915	5-acetyl-5,10-dihydrophenazine-1-carboxylic acid; 5-(2-hydroxyacetyl)-5,10-dihydrophenazine-1-carboxylic acid; endophenazine A1; endophenazine F; endophenazine G	30%
Region 2.3	Ni–siderophore	920,994–933,354	desferrioxamine E	75%
Region 3.1	linaridin	65,567–88,156	5-dimethylallylindole-3-acetonitrile	33%
Region 3.2	ectoine	149,753–160,127	kosinostatin	8%
Region 6.1	ectoine	13,107–23,508	ectoine	100%

**Table 2 microorganisms-11-02899-t002:** Differential characteristics of strain Q11^T^ compared to its relative reference strains in the genus *Flexivirga.* Strains: 1, Q11^T^; 2, *Flexivirga aerilata* ID2601S^T^ 3, *Flexivirga caeni* BO-16^T^; and 4, *Flexivirga endophytica* KCTC 39536^T^. The data of the reference strains are from previous studies. Symbols: +, positive; −, negative; and w, weak reaction.

Characteristic	1	2 ^a^	3 ^b^	4 ^c^
Gram stain	positive	positive	positive	positive
Temperature range for growth (°C)	25–40	25–45	10–40	20–45
Optimal growth (°C)	30	35	25–37	28–35
pH range for growth	6.0–8.0	6.0–8.0	5.0–10.0	5.0–8.0
Optimal pH	7.0	7.0	7.0	7.0
Growth in NaCl (% *w*/*v*)	0–4	0–6	0–5	0–7
Optimal NaCl concentration (% *w*/*v*)	1	0	1	0–3
Tween 60	+	−	−	+
Alkaline phosphatase	+	−	+	+
Esterase (C4)	−	+	+	+
Esterase Lipase (C8)	w	−	+	+
Valine arylamidase	w	w	+	+
Cystine arylamidase	w	w	+	+
*α*−Galactosidase	+	−	+	−
*β*−Galactosidase	+	−	+	−
*β*−Glucosidase	+	+	−	+
*N*−Acetyl−*β*−glucosaminidase	+	−	−	+
*α*−Fucosidase	+	−	+	−
Gelatinase	+	−	−	−
*β*−Galactosidase	+	−	+	−
Glucose	+	+	−	−
Mannose	+	−	+	−
Mannitol	−	+	−	+
*N*−Acetyl−glucosamine	+	−	+	−
Malic acid	+	+	−	−
Citrate	+	+	−	+
Phenylacetic acid	−	−	+	+
Erythritol	+	− −	−	−
d−Galactose	+	−	−	−
d−Fructose	+	−	−	+
DNA G + C content (mol%) *	68.5	69.8	68.0	66.7

^a^, Data from [6]; ^b^, data from [5]; ^c^, data from [2]. * DNA G+C content (mol%) was obtained from genomic data.

**Table 3 microorganisms-11-02899-t003:** Cellular fatty acid content of strain Q11^T^ and its relative reference strains in the genus *Flexivirga.* Strains: 1, Q11^T^; 2, *Flexivirga aerilata* ID2601S^T^ 3, *Flexivirga caeni* BO-16^T^; and 4, *Flexivirga endophytica* KCTC 39536^T^. The data of the reference strains are from previous studies. Values are percentages of total fatty acids. Fatty acids that represented <0.5% in all strains were omitted. TR, trace amount (<0.5%); -, not detected. Summed features are groups of two or more fatty acids that cannot be separated using the MIDI system. Summed feature 3 contains C_16:1_ *ω*6c and/or C_16:1_ *ω*7c, and summed feature 9 contains C_16:0_ 10-methyl and/or iso-C_17:1_
*ω*9c.

Fatty Acids	1	2 ^a^	3 ^b^	4 ^c^
C_16:0_	1.0	0.77	1.1	1.4
C_16:1_ 2OH	-	1.5	5.9	TR
C_18:0_	7.3	1.6	-	-
C_18:0_ 10-methyl	7.0	2.8	TR	0.6
C_18:1_ *ω*9c	3.5	1.8	-	0.8
C_19:0_ cyclo *ω*8c	1.0		-	-
iso-C_15:0_	1.9	2.4	4.6	4.5
iso-C_16:0_	19.7	24.6	39.8	27.6
iso-C_16:1_ H	2.3	3.4	0.6	3.2
iso-C_17:0_	10.5	8.3	9.0	9.8
iso-C_18:0_	4.9	3.5	1.4	1.9
iso-C_19:0_	1.0	-	-	-
anteiso-C_15:0_	tr	-	1.0	1.8
anteiso-C_17:0_	8.7	10.6	24.3	23.5
anteiso-C_17:1_ *ω*9c	3.1	4.5	0.8	9.1
anteiso-C_19:0_	1.6	0.9	TR	0.5
Summed feature 3	1.7	1.7	-	2.7
Summed feature 9	22.5	17.2	4.0	8.5

^a^, Data from [6]; ^b^, data from [5]; ^c^, data from [2].

## Data Availability

The datasets generated and/or analyzed during the current study are available from the corresponding author upon reasonable request.

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
