# Peer review of "Endophytic Bacterium *Flexivirga meconopsidis* sp. nov. with Plant Growth-Promoting Function, Isolated from the Seeds of *Meconopsis integrifolia"

_microorganisms, 2023, doi:10.3390/microorganisms11122899_

Round 1

Reviewer 1 Report

Comments and Suggestions for Authors

Dear Authors!

The manuscript is devoted to the actual topic of identifying effective and versatile beneficial bacteria to improve plant growth and identification a new species of microorganisms. But in order to improve its quality, I propose to make some changes to it.

1. In the Introduction, it is necessary to write in more detail about what beneficial properties other bacteria of the genus Flexivirga have, and whether they belong to the PGPB group.

2. The Introduction should state the purpose of the pot experiments.

3. What database was used to identify fatty acids?

4. Section 2.6 should be written in more detail. How was proteomic analysis performed at Shanghai Paisenuo Biology Co., Ltd?

5. What chlorophyll content was determined (a, b or total)?

6. Line 270. Optimal growth temperature is 35oC. The abstract indicates a temperature of 30 oC.

7. Line 289. The species of plant is incorrectly specified.

8. Lines 303-306. “...this study provides certain insights for improving the growth and development of Meconopsis integrifolia, and complements the study of microbial and plant interaction in alpine ecosystems. What information and additions do the authors have in mind?

9. The Discussion section does not contain a discussion as such; it merely restates the results obtained by the Authors.

Comments on the Quality of English Language

Moderate editing of English language required

Reviewer 2 Report

Comments and Suggestions for Authors

This is a thorough characterization of a novel endophyte with interesting physiological properties.

Minor corrections suggested as follows:

102      2.6 Shutgun Proteomics Analyses  2.6 Shotgun Proteomics Analyses

118       Inoculate strain Q11T in LB  Strain Q11T was inoculated in LB

119       dilute  diluted

122      Sow the surface disinfect maize seeds  Surface disinfected maize seeds were sown

192      can protect  could protect

287      Interplant microorganisms  Endophytic microorganisms

289     Artemisia aerea  Artemisia aerea

Comments on the Quality of English Language

The quality of the English writing is very high, with a few exceptions noted above.

Reviewer 3 Report

Comments and Suggestions for Authors

Introduction

This part looks too short. I recommend the authors to expand the initial presentation of the positive properties of cells of genus Flexivirga, which were identified earlier in published studies, in order to strengthen the motivation in this article in favor of the relevance of the new studies undertaken by the authors of the article. The representation of the relevance of the work should be strengthened in Introduction.

Methods

Line 80: please, specify the certain compounds after the phrase “possible secondary metabolites [21].”

Lines 98-101: All methods should be briefly disclosed in the text instead of the short text like “Qualitative and quantitative analysis of cell fatty acids was performed by the method of Sakamoto et al. [28]. The respiratory quinone of strain Q11T was extracted by the method of Minnikin et al. [29], and analyzed by LC-MS. Polar lipids were extracted, separated, and identified by the described by Xu et al. [30].” Please check all parts of the text in Section 2 by the necessary additions of the methods’ explanation.

All types of equipment used by the authors to realize the methods should be given in the text.

Part 2.6: Please, add the reference to “shotgun proteomics” used in investigation.

Line 114: “inorganic” should be here instead of “norganic”

Line 123: please, specify the type and main characteristics of the soil used in pot experiments.

Results

Table 2: Please put “Optimal pH” and “Optimal NaCl concentration” instead of “Optimal” correspondently. Please, give the concentration dimension of NaCl, and concentrations of Tween 60 and other compounds used as tested substrates. All methods used to check different enzymatic activities and present in Table 2 should be performed in Methods.

Lines 250-251: There is the following sentence “The discovery of strain Q11T provides some basis for the related research of Meconopsis integrifolia.” It is necessary to add a special text here which can help to construct some logic transition from of Meconopsis integrifolia to corn. The beginning of the next section with the investigation of corn after the mentioned sentence looks absolutely unclear.

Discussion

I recommend the authors to add the comparison of the properties of isolated and studied new strain with previously known strains of cells of genus Flexivirga. Does the new strain possess some very interesting properties which are absent in other known strains? Such accent in discussion can increase the importance of new discovered strain.

Conclusion

It is absent as a separate section in the text, but it should be added to the article, the actuality and novelty of the obtained results, as well as perspectives of the new strain use should be presented here.

The phrase “In conclusion, this study provides certain insights for improving the growth and development of Meconopsis integrifolia, and complements the study of microbial and plant interaction in alpine ecosystems.” has no direct factual results in the article and experiments with plants were carried out with corn. So, this part of the text should be modified.

References

Please, check the information in all references, since there are some unclear things and doublings:

Reference # 1: The Journal of Antibiotics 2011 64:9 2011, 64, 613–616

Reference #4: Hyeon, J.W.; Kim, H.R.; Lee, H.J.; Jeong, S.E.; Choi, S.H.; Jeon, C.O. Flexivirga Oryzae Sp. Nov., Isolated from Soil of a Rice 348 Paddy, and Emended Description of the Genus Flexivirga Anzai et al. 2012. Int J Syst Evol Microbiol 2017, 67, 479–484.

Reference #36: Microorganisms 2023, Vol. 11, Page 2317 2023, 11, 2317

Reference #44: Nature Reviews Microbiology 2018 16:5 2018, 16, 304–315

Reference #45: Molecules 2020, Vol. 25, Page 1032 2020, 25, 1032

Reference #47: Marine Drugs 2019, Vol. 17, Page 127 2019, 17, 127

All authors should be mentioned in the references instead of “et al.”

The References #5 and #7 correspond to the same paper.

The References #9 and #35 correspond to the same paper.

Generally:

It is not necessary to indicate the authors of the references used in the text; it is enough to indicate their numbers. Please, see the following lines: 35 ( Anzai et al [1]), 57 (Liu et al [11]), 72-73 (Sun et al. [18]), 84 (Halebian et al [23]), 89 (Wang et al [26]), 99 (Sakamoto et al. [28]), 100 (Minnikin et al. [29]), 101 (Xu et al. [30]), 109 (Hamane et al. [31]), 206 (Chaudhary et al [6], Keum et al [7],   from Gao et al [2].)

Please, put the data of the last usage for all databases, websites and on-line programs mentioned in the text.

All types of equipment used must be specified correctly (type, number, company, country) so that readers can think about the possible reproducibility of the results obtained.

Round 2

Reviewer 3 Report

Comments and Suggestions for Authors

The authors have noticeably improved the text by making additions, but I still have a few minor recommendations for them:

Line 92-93: There is no verb in the sentence.

Line 157-159: Please, delete the sentence “The SPAD-502 Plus determines the 157 relative chlorophyll content in leaves by measuring the difference in optical concentration of the leaves at two wavelengths (650 nm and 940 nm) [47]”. It is incorrect, especially the phrase optical concentration”. According to the reference [47} the intensity of fluorescence of chlorophyll was measured after dark and Light exposition. The extract of chlorophyll was spectrophotometrically measured at 480, 649 and 665 nm. What did the authors of the current paper measure at 940 nm? Please, add explanations and see the instruction to the used chlorophyll meter (Konica Minolta SPAD-502 Plus, Japan).

Table S5: “Physicochemical” should be instead of “Physiochemical”.

Figure 3H: “Chlorophyll” should be instead of “Chlorophyl”.

It is necessary to check the spelling of the text (including Supplementary materials) and rephrase the “heavy” sentences.  For example, I recommend the sentence “Microorganisms use iron carriers to provide plants with this metal, which stimulates growth and increases plant resistance to stress” instead of “Microorganisms use iron carriers to provide iron to plants to promote growth and improve plant stress resistance” (lines 357-358).

Comments on the Quality of English Language

It is necessary to check the spelling of the text (including Supplementary materials) and rephrase the “heavy” sentences. 
